# Cornelian Cherry (*Cornus mas*) Powder as a Functional Ingredient for the Formulation of Bread Loaves: Physical Properties, Nutritional Value, Phytochemical Composition, and Sensory Attributes

**DOI:** 10.3390/foods12030593

**Published:** 2023-01-31

**Authors:** Veronika Šimora, Hana Ďúranová, Ján Brindza, Marvin Moncada, Eva Ivanišová, Patrícia Joanidis, Dušan Straka, Lucia Gabríny, Miroslava Kačániová

**Affiliations:** 1AgroBioTech Research Centre, Slovak University of Agriculture, Trieda Andreja Hlinku 2, 94976 Nitra, Slovakia; 2Institute of Horticulture, Faculty of Horticulture and Landscape Engineering, Slovak University of Agriculture, Trieda Andreja Hlinku 2, 94976 Nitra, Slovakia; 3Institute of Plant and Environmental Sciences, Slovak University of Agriculture, Trieda Andreja Hlinku 2, 94976 Nitra, Slovakia; 4Department of Food, Bioprocessing, and Nutrition Science, North Carolina State University, Raleigh, NC 27606, USA; 5Institute of Food Sciences, Slovak University of Agriculture, Trieda Andreja Hlinku 2, 94976 Nitra, Slovakia; 6Department of Bioenergy, Food Technology and Microbiology, Institute of Food Technology and Nutrition, University of Rzeszow, 4 Zelwerowicza Str., 35-601 Rzeszow, Poland

**Keywords:** bakery, biologically active substances, DPPH assay, nutritional value, sensory properties

## Abstract

In the current study, Cornelian cherry powder (CCP, *Cornus mas*) was investigated as a functional ingredient for bread production. Experimental bread loaves were prepared using five levels of CCP (0, 1, 2, 5, and 10% *w*/*w*) to replace wheat flour in bread formulation. The final products were analyzed regarding their proximate composition, content of selected biologically active substances, antioxidant activity (AA), volume, and sensory attributes. Increasing the incorporation of CCP led to significantly (*p* < 0.05) higher concentrations of carbohydrate, ash, energetic value, total polyphenols, phenolic acids and AA, and reduced fat and protein contents (*p* < 0.05). Moreover, up to 5% addition of CCP positively affected the volume (642.63 ± 7.24 mL) and specific volume (2.83 ± 0.02 cm^3^/g) of bread loaves, which were significantly (*p* < 0.05) higher compared to the control (no addition of CCP; 576.99 ± 2.97 mL; 2.55 ± 0.002 cm^3^/g). The sensory attributes chewiness, crumb springiness, bitterness, and sourness had lower scores (*p* < 0.05) in bread formulated with 10% CCP compared to the control. Overall, results show that the bread loaves produced with up to 5% CCP addition were considered the preferred formulation among the experimental samples tested, taking into consideration their composition, bioactive content, sensory, and physical properties.

## 1. Introduction

At a global level, bakery goods constitute an essential part of human nutrition [1]. The development of innovative products using value-added ingredients has become an important trend in the bread manufacturing industry in an effort to meet the demand of a new generation of consumers seeking healthier lifestyles [2]. Fruits are a remarkable source of natural bioactive compounds with great potential of incorporation in the formulation of multiple products [3,4,5]. However, fresh fruits are highly perishable due to their extremely high moisture content (MC; 75–95%) [6], which leads to estimated post-harvest losses of around 30% [7]. Therefore, efficient preservation techniques are necessary to extend their shelf-life and marketability [8]. In this regard, drying technology has proven to be a feasible, convenient, and well-accepted processing strategy for improving food storage stability [9,10]. Several types of dehydration technologies are available, e.g., solar drying, hot air drying, vacuum drying, spray drying, osmotic drying, microwave drying, and freeze drying [11]. Among them, hot air drying is currently the most widely used method in the agri-food sphere. Indeed, a more homogeneous, sanitary, and coloured dehydrated product can be quickly obtained using this process [12]. Ultimately, powdered dried fruits provide the producers with enhanced nutritional and health properties for the many types of cereal products [13]. In effect, the addition of powdered fruits to the recipe of the bakeware offers an important protective capacity against a number of civilization disorders due to their antioxidant, anti-inflammatory, anti-mutagenic, and anti-carcinogenic characteristics [14]. For example, enriched foods showed enhanced antioxidant power, which can help prevent some chronic diseases [15]. 

In this regard, many fruit-derived materials have been examined, including mango peel [4], banana [3], pomegranate peel [16], orange peel [17], and apple pomace [18]. From a technologically advanced point of view, fruit-derived ingredients are naturally gluten-free, and this might modify important attributes of the flour blend and consequently the properties of the final bread [19]. For instance, decreased volume and cohesiveness of bread loaves formulated with mango peel powder (more than 5% *w/w*) addition has been noted by Chen et al. [4]. Moreover, decreased volume, specific volume, moisture, elasticity, and cohesiveness of bread loaves with increasing concentrations of pomegranate peel powder (0%, 1%, 3%, 5%, and 7%) have been found by Zhang et al. [16]. On the other hand, sensory ratings shown by Baba et al. [3] revealed no significant differences in taste, aroma, and appearance of wheat bread containing banana powder (up to 30%) as compared to the control sample, but the bread with the 30% addition had lower overall acceptability. Hence, the careful selection of ingredients and the determination of their adequate concentration [20] are of great importance for a successful food practice.

Cornelian cherries (CC; *Cornus mas* L.) are currently gaining increasing attention from the scientific research community [21]. These oval or pear-shaped edible fruits with color ranging from red to purple constitute an important source of vitamin C and polyphenols, mainly flavonoids, anthocyanins, and iridoids [21,22,23]. Such substances are linked to a wide range of biological effects and pharmacological properties, including antimicrobial, anti-inflammatory, anti-cancer, anti-diabetic, and anti-atherosclerotic activities [24,25,26]. Most often, the cherries are consumed fresh or as a dried delicacy. Due to their beneficial properties, they can serve as purposeful ingredients for the commercial food sector [27]. In recent years, CC has also been applied as a flavoring ingredient in ice creams, desserts, and cakes [28]. Furthermore, CC and their juices or extracts have inspired the production of novel foods, such as beer [29], soup [30], vinegar [31], or burgers [32]. Regarding baked goods, İlyasoğlu et al. [33] replaced wheat flour (WF) with CC (10 g 100 g^−1^ composite flour) in cookie formulations and reported enhanced contents of omega-3 fatty acids, total phenolic content, and antioxidant capacity in the final product. 

To the best of our knowledge, the addition of Cornelian cherry powder (CCP) to bread formulation has not yet been investigated. Therefore, this study is the first report showing the production and quality assessment of bread loaves produced with CCP. For this, the incorporation of different CCP concentrations (1, 2, 5, and 10% *w*/*w*) for the partial substitution of WF in the formulation of bread loaves was performed, and the quality characteristics of the final products were determined. To achieve our goal, we assessed the compositional profile, antioxidant activity (AA), phenolic compounds, key physical attributes, and sensory properties of bread loaves. As a result, recommendations about the most appropriate level of CCP for bread production are discussed. This report unveils the potentiality of Cornelian cherry powder as a functional ingredient for bakery products.

## 2. Materials and Methods

### 2.1. Materials

Wheat flour (WF; T-650 type) was purchased from a grinding mill (Pohronský Ruskov JSC, Pohronský Ruskov, Slovakia). Other raw materials for bread making, such as salt (Solivary Trade Ltd., Trenčín, Slovakia), saccharose (Slovenské Cukrovary Ltd., Sereď, Slovakia), and compressed yeast (Thymos Ltd., Veľká Lomnica, Slovakia), were obtained from a local grocery store. The chemical reagents were of analytical quality, and all of them were purchased from Reachem (Bratislava, Slovakia) or Sigma Aldrich (Saint Louis, MO, USA).

### 2.2. Preparation of Cornelian Cherry Powder (CCP)

Fresh, fully ripe Cornelian cherry (CC; *Cornus mas*) fruits were collected from the SUA Botanical Garden (Slovak University of Agriculture, SUA, Nitra, Slovakia) and subsequently selected, cleaned, and pitted. For the preparation of Cornelian cherry powder (CCP), the fruits were dried at 45 °C until complete dehydration using a cabinet dryer (Universal oven UF 160, Memmert GmbH + Co.KG, Büchenbach, Germany). After this, the dried CC was homogenized (ETA Gratus 0028 90030, ETA-Slovakia Ltd., Bratislava, Slovakia) and sieved to obtain powder particles with a diamter of 0.5 mm. The produced CCP was packed in polyethylene (PE) bags and stored at room temperature in a dark place until analysis and/or use. 

### 2.3. Composite Flour Preparation

Five composite flours were prepared by partially replacing WF with CCP according to the following ratios: WF:CCP, 100:0 (control sample), 99:1 (1% addition of CCP), 98:2 (2% addition of CCP), 95:5 (5% addition of CCP), and 90:10 (10% addition of CCP), respectively. The WF/CCP blends were individually packaged in PE bags and kept at room temperature until the bread making process. 

### 2.4. Bread Loaf Preparation

Bread loaf experimental treatments (Figure 1) were prepared following the methodology described by Valková et al. [18]. The bread formulation consisted of 500 g of WF or WF/CCP blends, saccharose (1% of flour), salt (2% of flour), water (60% of flour), and yeast (2% of flour). Initially, compressed yeasts were reactivated in a saccharose solution at 32 °C for 5 min. All ingredients were blended for 10 min in two steps (first step: 4 min at 1500 rpm; second step: 6 min at 3000 rpm) in a mixer (DIOSNA SP 12; DIOSNA Dierks and Söhne GmbH, Osnabrück, Germany) using a dough hook accessory to ensure proper hydration of flour. Then, the dough was carefully portioned into 250 ± 5 g pieces and placed into oiled and floured tins. The tins were transferred to a fermentation cabinet (MIWE cube, Pekass Ltd., Pilsen, Czech Republic) set at 32 °C and 85% relative humidity, and allowed to proof for 40 min. The bread loaves were baked in two phases (Phase I: 180 °C with the addition of 160 mL steam at the same temperature; Phase II: 210 °C for 7 min, no steam) in a steamer oven (Laboratory oven MIWE cube, Pekass Ltd., Pilsen, Czech Republic). The baked loaves were removed from the tins and left to cool at room temperature for 2 h until cutting. In total, three batches of each type of bread were produced (15 bread loaves in total). 

### 2.5. Determination of Compositional Profile and Energetic Value 

The compositional value of CCP and experimental bread loaves was assessed. For this purpose, moisture, ash, crude protein, fat, and total carbohydrate contents, as well as the energetic values, were determined. 

The moisture content (MC) was measured with an automatic moisture analyzer, DBS 60-3 (Kern and Sohn GmbH, Altstadt, Germany), according to the manufacturer’s instructions and the ASTM D 6980 method. Briefly, 1 g of sample was placed on the sample plate and tested at 120 °C for the required time (10–15 min). The total ash and crude protein contents were determined in accordance with AACC standard 08-01 using a muffle furnace (Neberterm, Germany) and the semi-micro Kjeldahl method (factor of converted nitrogen to protein was 6.25). The total fat content was analyzed using the Ankom XT15 Fat Extractor (Ankom Technology, Fairport, NY, USA) in accordance with the manufacturer’s instructions. The total carbohydrate content (TCC) and energy were calculated by Equations (1) and (2), according to Valková et al. [18] and Arraibi et al. [34]:TCC (%) = 100 − moisture (%) − protein (%) − lipids (%) − ash (%)(1)
Energy (kcal/100 g) = 4 × (% proteins + % carbohydrates) + 9 × (% fat)(2)

### 2.6. Determination of Radical Scavenging Activity and Polyphenolic Compounds 

Firstly, samples of ethanolic extracts (CCP, and bread treatments) were prepared. For each extraction, 0.2 g (CCP) and 0.5 g (bread treatments) of sample received 20 mL or 40 mL of 80% ethanol, respectively, and were extracted for 2 h, followed by centrifugation at 4000× *g* for 10 min in a Rotofix 32A (Hettich, Spenge, Germany). The supernatant was used for the determination of antioxidant activity (AA), total polyphenols content (TPC), total phenolic acids content (TPAC), and flavonoid content (FC).

The AA, TPC, TPAC, and FC of the samples were analyzed using the 2,2-diphenyl-1-picrylhydrazyl (DPPH) assay [18], the colorimetric assay utilizing the Folin-Ciocalteu (F-C) reagent [18], and according to the procedures of Valková et al. [18] and Ivanišová et al. [35], respectively. The AA was expressed as Trolox equivalent antioxidant capacity (TEAC) in milligrams per gram of dry weight (dw). Gallic acid (for TPC analysis), caffeic acid (for TPAC analysis), and quercetin (for FC analysis) standards were used, and the results were expressed as gallic acid equivalents (GAE), caffeic acid equivalents (CAE), and quercetin equivalents (QE) in milligrams per gram of dw, respectively.

### 2.7. Volume Analysis

The volume (mL) and specific volume (cm^3^/g) of bread samples were assessed using an automatically laser-based scanning device, the VolScan Profiler VSP 300 (Stable Micro Systems, Godalming, UK), according to the manufacturer’s recommendation (AACC approved method 10.16.01).

### 2.8. Sensory Assessment 

Sensory analysis was performed by 10 panelists (three men and seven women, aged 26–47), trained according to the standard STN EN ISO 8586. The evaluation of bread samples took place at the Sensory Laboratory of the Research Centre AgroBioTech (Slovak University of Agriculture, SUA, Nitra, Slovakia) during the late morning. The experimental samples (control and four treatments, 1, 2, 5, and 10% (*w*/*w*)) were coded with 3-digit numbers and presented to the panelists at the same time. Between evaluations of individual samples, the panelists were instructed to drink water.

Eleven sensory descriptors were evaluated (on a 15-point unstructured scale) and divided into the following categories: color (crust color, crumb color), texture (pore uniformity, crumb springiness, chewiness), aroma, taste (sweet taste, bitter taste, sour taste, aftertaste), and overall impression. The descriptors for sensory rating were selected according to García-Gómez et al. [36], and based on preliminary training sessions with selected panelists. 

### 2.9. Statistical Analysis

All the analyses were conducted at least in triplicate, and the data was reported as mean value ± standard deviation. One-way analysis of variance (ANOVA) and the Tukey test (Prism 8.0.1 program, GraphPad Software, San Diego, CA, USA) were applied to establish statistically significant differences between the samples at the level of *p* < 0.05.

## 3. Results

### 3.1. Characterization of Cornelian Cherry Powder

The compositional profile, energetic value, polyphenolic compounds, and AA of CCP are shown in Table 1. CCP had a low energetic value, and low fat and protein contents, but a high total carbohydrate and ash contents. Regarding DPPH free radical scavenging activity and polyphenolic compounds, the CCP exhibited high AA, and high total polyphenols, total phenolic acids, and flavonoid contents. 

### 3.2. Compositional Profile and Energetic Value of Experimental Bread Loaves

Increasing the percentage of CCP in the bread formulations led to a progressive and significant (*p* < 0.05) increase in total carbohydrate and ash contents, and energetic value, as shown in Table 2. Interestingly, no differences in total carbohydrates and energetic value were noted between bread loaves with 5% and 10% of CCP. In addition, a significantly (*p* < 0.05) linear reduction in fat and protein contents was observed when more CCP was added to the formulation of bread loaves (Table 2), which agrees with the low fat and protein contents found for CCP (Table 1). No significant differences were observed for the MC of bread treatments (*p* > 0.05).

### 3.3. Antioxidant Activity and Polyphenolic Compounds of Experimental Bread Loaves

The AA and content of selected polyphenolic compounds in experimental bread loaves enriched with CCP are summarized in Table 3. For all analyzed parameters, a significant and linear increase in bioactivity was demonstrated as higher ratios of CCP were used for the production of bread loaves (*p* < 0.05), with the exception of the 1% CCP treatment, which had similar results compared to the control sample (*p* > 0.05). Moreover, flavonoid compounds were not detected in any of the bread treatments (Table 3).

### 3.4. Volume of Experimental Bread Loaves

Breads produced with WF/CCP blends of 1%, 2%, and 5% CCP showed significantly different (*p* < 0.05) volume and specific volume (Table 4). The addition of CCP led to a significant increase in both parameters, volume and specific volume, which is a desirable attribute for bread loaves. However, bread loaves prepared with 10% CCP had significantly (*p* < 0.05) lower volume and specific volume compared to all treatments and the control sample. The highest results were observed for 2% CCP, which resulted in an increase of 15.8% and 16.4% for volume and specific volume, respectively. 

### 3.5. Sensory Properties of Experimental Bread Loaves

The crust and crumb colors of bread loaves formulated with ≥2% and ≥5% CCP, respectively, were reported as significantly (*p* < 0.05) darker compared to the control (Table 5). Significant differences in the bread aroma were identified between the control sample and the bread enriched with 2% and 10% CCP, respectively. Further, the evaluators perceived an aftertaste (*p* < 0.05) in samples produced with ≥1% CCP, while the highest sour and bitter taste scores (*p* < 0.05) were reported for bread produced with the highest (10%) CCP ratio. Additionally, this sample was reported as the least chewable and having poor crumb springiness. Regarding the pore uniformity and sweet taste parameters, there were no significant differences between experimental samples. When evaluating the overall impression, the sample with the 2% addition was perceived as the tastiest and had the highest score, which is significantly superior to the control (*p* < 0.05).

## 4. Discussion

In general, the functional characteristics of a raw material affect its interaction with other food components and strongly determine its final application [37]. Wheat flour is used as a major staple raw material in bread production [38], and it is the most abundant source of calories and protein in the human diet [39]. Although it is also a great source of nutrients, its content of bioactive compounds and AA is poor as a consequence of the refining during processing [40]. To improve the nutritional profile and the biological activity of bakery products, the partial replacement of WF with phytochemical-rich, functional plant-based flours or powders is an interesting strategy. In this sense, horticultural crops are known to provide a rich source of diverse nutritional molecules, many of them possessing antioxidant activity, which has been reported as capable of protecting the human body against oxidative cellular reactions [41]. Cornelian cherry-derived ingredients have been tested as part of several formulations to produce enhanced food products [28,32,33,42,43,44,45,46,47,48,49,50,51,52], but not in bread formulations yet. Therefore, this is the first report of such research activities with solid potential for practical applications to bread manufacturing. 

Regarding the characterization of CCP, Tontul et al. [53] recorded similar values for MC (8.03 ± 0.14%) in CCP dried at 50 °C. Since moisture in sugar-rich powders acts as a plasticizer [54], the lower MC of CCP used in our study may have a positive effect on its cohesive properties. Importantly, foods with reduced MC are considered safe due to the growth mitigation of undesirable microorganisms (especially molds), thus improving the shelf-life of the product [55]. We found CCP to have a higher ash content compared to dried CC fruits (2.83 ± 0.35%), as demonstrated by Petkova and Ognyanov [56]. Further, total carbohydrates represent the most essential source of energetic value in CC fruits [57], which is also in line with our findings. Likewise, relatively low crude protein levels (ranging from 1.43 to 2.71%) have also been demonstrated in CC fruits by Serbia by Bijelić et al. [58]. In fact, raw CC fruits typically have low fat content (1.49 ± 0.02%) [59], which was also confirmed by our results. Differences between our results and those of cited research studies can be explained by different genotypes used, as well as the influence of environmental growth conditions [60]. 

It is important to characterize not only the overall AA but also the individual antioxidant components responsible for such activity, which are present in diverse fruits [61,62,63,64]. Our findings demonstrate that oven-dried CCP produced in this study has high contents of total polyphenols, phenolic acids, and flavonoids along with a strong AA and reiterate results found in fresh CC fruits determined by Dupak et al. [65] and Szczepaniak et al. [66]. In contrast, Popović et al. [67] have identified a lower content of total polyphenols in dried samples of 10 CC genotypes. In addition, AA and the contents of total phenolic acids and flavonoids were lower in CC pulp analyzed by Dupak et al. [65] in comparison with our CCP. In fact, these discrepancies are expected since different processing protocols and parameters, fruit genotypes and varieties, and maturity stages affect the aforementioned results [25,68]. Considering our results, it can be hypothetically assumed that eating both our CCP and/or products enriched with the CCP could be beneficial for human health in the sense of their ability to eliminate harmful oxidative stress in the organism, thus reducing the risk of chronic disease incidence. 

The effect of CCP addition on the nutritional composition and key quality attributes of bread loaves enriched with four concentrations (1%, 2%, 5%, and 10%) of CCP was further evaluated. Carbohydrates are the prime macronutrients in bread. The content of carbohydrates in the bread formulations increased linearly with the incorporation of CCP, because CCP is also a major source of this macronutrient (>85%, Table 1). In addition, increasing additions of CCP led to progressively lower protein and fat contents and higher ash contents in the enriched bread loaves, also as a reflex of the original CCP composition (Table 1). The same trend was noted by Topdaş et al. [28], who analyzed the impact of different CC fruit paste (5%, 10%, and 15%) additions to the composition of ice cream. Similarly, a lower fat content in CCP-enriched biscuits compared to control samples was reported by İlyasoğlu et al. [33]. Given these findings, CCP may be a promising ingredient in the preparation of low-fat goods for the food sector. 

Moreover, our results pointed to markedly higher concentrations of phenolic compounds (TPC and TPAC) and AA in the CCP-containing bread loaves in comparison to the control. This finding agrees with the research conducted by İlyasoğlu et al. [33], Topdaş et al. [28], and Haghani et al. [51], which revealed increased levels of TPC and higher AA, as well as higher concentrations of CC in products used to prepare biscuits, conventional ice cream, and probiotic ice cream, respectively. Enhanced TPC and antioxidant capacity of white chocolate and dairy desserts after addition of CCP and CC juice, respectively, were also observed in the studies performed by Cerit et al. [45] and Ivanova et al. [47]. On the other hand, the total absence of FC identified in all our bread samples may be related to the thermolability of these biologically active substances [69] and possible complete degradation during the baking process. Indeed, flavonoids are major phenolic compounds with natural antioxidant capacity (mediated via their functional hydroxyl groups in their structure) [70], reported in CC fruits [63] and also here in our developed CCP (Table 1). The microencapsulation technique using appropriate wall materials has been reported in the literature [71] as an efficient stabilization approach for the preservation of polyphenolic extracts from Cornelian cherries [47]. Therefore, the spray-drying and microencapsulation of plant extracts to produce powdered ingredients with preserved biologically active compounds destined for bakery products will be considered in future studies by our research team. 

The addition of non-traditional ingredients to the bakery goods not only affects their nutritional composition and bioactivity properties but may also influence important physical attributes. One of the most crucial physical properties of bread is its volume, which strongly determines the consumers’ preferences and predicts its quality, as well [72]. In our study, bread samples enriched with up to 5% CCP displayed significantly higher values for volume and specific volume compared to control wheat bread. We hypothesize that this increase may be related to the presence of pectin and other hydrocolloids (estimated to be about 5.7% in dried CC fruits) in the CCP composition [73]. Confirming our hypothesis, Rosell et al. [74] reported that mixing WF with hydrocolloids increases dough stability and loaf volume due to their ability to absorb water and gelling properties, as well. In view of this, during heating a gel network may be formed, which can consequently strengthen the expanding dough cells, thereby improving gas retention and bread volume [75]. In addition, Das et al. [76] found that hydrocolloids prevent the small cells found in bread dough from clumping together to generate larger cells. Having a larger number of small cells can form a more uniform matrix that acts as a CO_2_-trapping network. Hydrocolloids improving the volume of bread loaves were also recognized in the research conducted by Kang et al. [77] and Zhao et al. [78]. Conversely, the reduction in volume and specific volume observed in our bread samples supplemented with 10% CCP may be attributed to an undesirably higher content of hydrophilic compounds (including carbohydrates) in these bread loaves, which could theoretically cause excessively higher viscosity. Furthermore, the reduced volume and increased stiffness of bread loaves enriched with higher CCP addition may also be due to a relevant reduction in the amount of gluten in the doughs [79], caused by the significant depletion of WF content in that formulation. Overall, our findings suggest that the addition of CCP up to 5% is a promising strategy to produce bread loaves and other related products with enhanced bioactivity and preserved bread quality parameters.

The sensory properties of a food product play a major role in its consumer acceptance and marketability [80]. When dealing with the incorporation of alternative ingredients into food formulations, the goal is to enhance the nutritional, bioactive, and physical attributes of the product without compromising its sensory acceptability [81]. In effect, higher levels of non-traditional bakery ingredients in bread formulations can greatly affect their taste and aroma [15], which are considered important sensory characteristics along with the texture [82]. Fresh, mature CC fruits have an intrinsic cherry-like, tart-sweet, and sour flavor with characteristic aroma [27,63] which might interfere with the sensory perception of a final product. Indeed, a gradual increase of aroma and taste (sweet, bitter, sour taste, aftertaste) scores of bread samples with the increment of CCP incorporation ratio was observed. Lowest scores for bitter, sour, and aftertaste were observed for bread formulated with 1% and 2% CCP addition. We hypothesize that bioactive compounds such as phenolics and iridoids found in CCP, which display many biological activities, such as anti-inflammatory, antioxidative, anti-cancer, anti-atherogenic, antidiabetic, and neuroprotective attributes [83], may play an important role in the observed sensory findings [84,85]. Further, as the CCP ratio increased, the crumb and crust color of CCP-enriched samples became darker (Figure 1). We suppose that this color modification could affect consumers’ acceptability in a positive manner. In effect, previously it was shown that the red color of bread (caused by red beetroot addition) was preferred by consumers as compared to that supplemented only with white beetroot [86]. In addition to these observations, the texture of our bread loaves was also modified by CCP addition. The increased chewiness of bread was positively correlated with the amount of CCP (10% addition), which is consistent with a previous study [87] documenting the superior chewiness of bread samples enriched with artichoke fiber. Similarly, the significant difference in springiness identified only between the bread with 10% CCP addition and the control reflects a dose-dependent effect of CCP addition on this specific parameter. Finally, the best overall impression score was reported for samples with 2% CCP addition, and it was significantly higher compared to the control, whereas the other treatments showed similar results (*p* > 0.05). Altogether, bread loaves prepared with 2% or 5% replacement ratios of WF with CCP proceed to be promising bakery products with desirable overall characteristics. Our results present a novel, functional approach for the development of wheat bread with enhanced attributes for the current health-oriented bakery market. 

## 5. Conclusions

Our study investigated the production and application of Cornelian cherry powder as a potential functional food ingredient for the partial replacement of WF in bread formulations. Our results show that the incorporation of CCP (replacement ratios between 1–10% *w*/*w*) in wheat bread formulations produces final products with significantly different composition, bioactivity, volume, and sensory attributes compared to the control. Indeed, increasing ratios of CCP lead to bread loaves with significantly higher (*p* < 0.05) carbohydrate and ash contents, energetic value, TPC, TPAC, and AA, but lower fat and protein contents. Further, our findings showed that CCP added at 1% to 5% ratios significantly (*p* < 0.05) improved the volume and specific volume of experimental bread loaves, and the highest overall impression score, significantly higher compared to the control, was reported for samples with 2% CCP addition. Overall, here we demonstrate that CCP can partially replace WF when used up to 5% (*w*/*w*) for bread formulations without negatively impacting key physical properties and sensory attributes, while enhancing the concentration of phenolic antioxidants. Moreover, we believe that the incorporation strategy shown here can be successfully applied to multiple bakery products for the production of healthier and more functional food products for the emerging health-oriented market.

## Figures and Tables

**Figure 1 foods-12-00593-f001:**
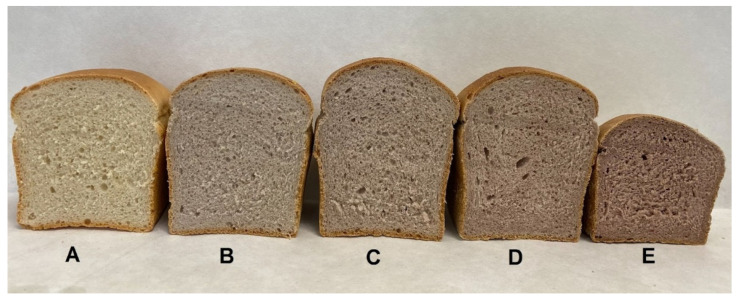
Experimental bread loaves. (**A**) control sample, (**B**) 1% Cornelian cherry powder addition, (**C**) 2% Cornelian cherry powder addition, (**D**) 5% Cornelian cherry powder addition, and (E) 10% Cornelian cherry powder addition.

**Table 1 foods-12-00593-t001:** Composition, energetic value, antioxidant activity, and polyphenolic compounds of Cornelian cherry powder.

Parameters	CCP
Fat (%)	0.20 ± 0.03
Carbohydrate (%)	86.88 ± 0.93
Protein (%)	0.78 ± 0.05
Ash (%)	5.47 ± 0.10
Energetic value (kcal/100 g)	352.44 ± 3.67
Moisture (%)	6.67 ± 0.85
AA (mg TEAC/g)	8.75 ± 0.01
TPC (mg GAE/g)	9.08 ± 0.54
TPAC (mg CAE/g)	2.62 ± 0.15
FC (mg QE/g)	3.62 ± 0.30

Values are expressed as the mean ± standard deviation (*n* = 3). CCP—Cornelian cherry powder; AA—antioxidant activity expressed as mg of Trolox equivalents per gram dry weight; TPC—total polyphenols content expressed as mg of gallic acid equivalents per gram dry weight; TPAC—total phenolic acids content expressed as mg of caffeic acid equivalents per gram dry weight; FC—flavonoid content expressed as mg of quercetin equivalents per gram dry weight.

**Table 2 foods-12-00593-t002:** Compositional profiles of experimental bread loaves.

Parameters	Incorporation Ratio of CCP (% *w*/*w*) ^1^				
	0	1	2	5	10
Fat (%)	6.57 ± 0.05 ^a^	6.26 ± 0.06 ^b^	6.00 ± 0.04 ^c^	5.52 ± 0.06 ^c^	5.05 ± 0.08 ^d^
Carbohydrate (%)	69.58 ± 0.19 ^d^	70.05 ± 0.12 ^c^	70.68 ± 0.15 ^b^	71.65 ± 0.44 ^a^	72.45 ± 0.40 ^a^
Protein (%)	13.90 ± 0.08 ^a^	13.61 ± 0.06 ^b^	13.12 ± 0.03 ^c^	12.60 ± 0.08 ^d^	12.06 ± 0.06 ^e^
Ash (%)	0.62 ± 0.03 ^e^	0.70 ± 0.02 ^d^	0.75 ± 0.02 ^c^	0.87 ± 0.04 ^b^	0.99 ± 0.03 ^a^
Moisture (%)	9.32 ± 0.20	9.38 ± 0.16	9.46 ± 0.09	9.35 ± 0.46	9.45 ± 0.41
Energetic value (kcal/100 g)	393.08 ± 0.94 ^a^	391.01 ± 0.93 ^b^	389.16 ± 0.34 ^c^	386.71 ± 1.75 ^d^	383.47 ± 1.53 ^d^

^1^ Incorporation ratio of partial substitution of wheat flour; please see item 2.3 for further details. Values are expressed as the mean ± standard deviation (*n* = 3). Data in the same line with different superscript letters are significantly different (Tukey’s test, *p* < 0.05). CCP—Cornelian cherry powder.

**Table 3 foods-12-00593-t003:** Antioxidant activity and selected polyphenolic compounds of experimental bread loaves.

Parameters	Incorporation Ratio of CCP (% *w*/*w*) ^1^				
	0	1	2	5	10
AA (mg TEAC/g)	0.60 ± 0.01 ^d^	0.61 ± 0.04 ^d^	0.69 ± 0.03 ^c^	0.76 ± 0.02 ^b^	1.22 ± 0.02 ^a^
TPC (mg GAE/g)	3.82 ± 0.11 ^d^	3.98 ± 0.09 ^d^	4.59 ± 0.19 ^c^	4.90 ± 0.03 ^b^	5.12 ± 0.02 ^a^
TPAC (mg CAE/g)	0.49 ± 0.09 ^d^	0.55 ± 0.03 ^d^	0.68 ± 0.05 ^c^	0.77 ± 0.02 ^b^	1.18 ± 0.05 ^a^
FC (mg QE/g)	ND	ND	ND	ND	ND

^1^ Incorporation ratio of partial substitution of wheat flour; please see item 2.3 for further details. Values are expressed as the mean ± standard deviation (*n* = 3). Data in the same line with different superscript letters are significantly different (Tukey’s test, *p* < 0.05). CCP—Cornelian cherry powder; AA—antioxidant activity expressed as mg of Trolox equivalents per gram dry weight; TPC—total polyphenols content expressed as mg of gallic acid equivalents per gram dry weight; TPAC—total phenolic acids content expressed as mg of caffeic acid equivalents per gram dry weight; FC—flavonoid content expressed as mg of quercetin equivalents per gram dry weight. ND—not detected.

**Table 4 foods-12-00593-t004:** Volume of the experimental bread loaves.

Incorporation Ratio of CCP (% *w*/*w*) ^1^	Volume (mL)	Specific Volume (cm^3^/g)
0	576.99 ± 2.97 ^d^	2.55 ± 0.002 ^d^
1	604.38 ± 8.48 ^c^	2.69 ± 0.05 ^c^
2	668.17 ± 6.56 ^a^	2.97 ± 0.04 ^a^
5	642.63 ± 7.24 ^b^	2.83 ± 0.02 ^b^
10	443.63 ± 1.22 ^e^	1.94 ± 0.02 ^e^

^1^ Incorporation ratio of partial substitution of wheat flour; please see item 2.3 for further details. Values are expressed as the mean ± standard deviation (*n* = 3). Data in the same column with different superscript letters are significantly different (Tukey’s test, *p* < 0.05). CCP—Cornelian cherry powder.

**Table 5 foods-12-00593-t005:** Sensory analysis of the experimental bread loaves.

Parameters	Incorporation Ratio of CCP (% *w*/*w*) ^1^				
	0	1	2	5	10
Crust color	4.00 ± 0.00 ^b^	5.48 ± 1.00 ^cd^	7.03 ± 2.10 ^ad^	9.16 ± 2.34 ^a^	8.77 ± 2.73 ^ac^
Crumb color	2.50 ± 0.00 ^d^	5.98 ± 1.73 ^ce^	7.44 ± 1.67 ^be^	9.02 ± 1.76 ^ae^	11.52 ± 1.47 ^a^
Pore uniformity	8.00 ± 0.00 ^a^	8.48 ± 1.59 ^a^	7.85 ± 2.09 ^a^	8.38 ± 1.54 ^a^	9.79 ± 2.33 ^a^
Aroma	4.00 ± 0.00 ^b^	4.50 ± 0.65 ^ab^	5.69 ± 1.48 ^a^	6.14 ± 2.45 ^ab^	7.89 ± 3.44 ^a^
Crumb springiness	13.00 ± 0.00 ^a^	12.50 ± 1.36 ^a^	13.17 ± 2.02 ^a^	10.70 ± 3.09 ^ac^	7.92 ± 2.63 ^bc^
Chewiness	6.00 ± 0.00 ^bc^	6.17 ± 0.80 ^bc^	5.97 ± 1.09 ^bc^	6.64 ± 1.87 ^ac^	8.45 ± 1.37 ^a^
Sweet taste	3.00 ± 0.00 ^a^	3.50 ± 0.86 ^a^	3.68 ± 0.83 ^a^	3.65 ± 1.43 ^a^	4.04 ± 2.07 ^a^
Bitter taste	1.00 ± 0.00 ^bc^	1.21 ± 0.30 ^ac^	1.43 ± 0.80 ^ac^	2.47 ± 1.49 ^ac^	2.72 ± 1.70 ^a^
Sour taste	1.00 ± 0.00 ^c^	1.65 ± 0.85 ^bc^	1.44 ± 0.41 ^b^	2.56 ± 1.07 ^b^	6.62 ± 2.93 ^a^
Aftertaste	3.00 ± 0.00 ^b^	4.11 ± 0.96 ^a^	4.63 ± 1.36 ^a^	5.99 ± 1.81 ^a^	7.92 ± 2.88 ^a^
Overall impression	11.00 ± 0.00 ^bcde^	11.58 ± 1.37 ^ae^	12.12 ± 1.04 ^a^	10.38 ± 2.40 ^ac^	8.54 ± 2.69 ^ad^

^1^ Incorporation ratio of partial substitution of wheat flour; please see item 2.3 for further details. Values are expressed as the mean ± standard deviation (*n* = 3). Data in the same line with different superscript letters are significantly different (Tukey’s test, *p* < 0.05). CCP—Cornelian cherry powder.

## Data Availability

Not applicable.

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
