# Peer review of "Cornelian Cherry (Cornus mas) Powder as a Functional Ingredient for the Formulation of Bread Loaves: Physical Properties, Nutritional Value, Phytochemical Composition, and Sensory Attributes"

_foods, 2023, doi:10.3390/foods12030593_

Round 1

Reviewer 1 Report

How the amount of carbohydrates in the bread increase compared with the flour? It must have decreased with fermentation (Tables 1 and 3 73.57 and 77.22, respectively). Check values and calculations.
By increasing the available carbohydrates by adding CCP, there may have been more CO2 production, and the bread samples have higher volume than the control sample, so it is not necessarily the presence of pectins or hydrocolloids that is mentioned in lines 397 - 400. A positive control with more sugar (sucrose) would be desirable.
In lines 409 - 412 an effect on bread viscosity. It was not measured, so it could not be stated.
How the descriptor "overall impression" was defined? This must be specified, it is a descriptor that measures intensity or is evaluating preference. Using the same judges to describe the product and measure preference is not recommended. On the other hand, a lower value could have a higher consumer acceptance, as read in the conclusions (lines 456-457), is it correct?
Table 6 shows that the sample with a 10% addition of CCP has the highest pore uniformity. It does not correspond with what is observed in Figure 1.
In Table 1, the comparison of means is unnecessary since they are different products.
In addition, when the comparison of means is performed, it is usual that the same trend is assigned to the order of the letters. In all tables, there is no order of the letters assigned.

Author Response

Reviewer #1

Point 1: How the amount of carbohydrates in the bread increase compared with the flour? It must have decreased with fermentation (Tables 1 and 3 73.57 and 77.22, respectively). Check values and calculations.

Response: Thank you for the notice. The values have been checked and corrected directly in the manuscript.

Point 2: By increasing the available carbohydrates by adding CCP, there may have been more CO2 production, and the bread samples have higher volume than the control sample, so it is not necessarily the presence of pectins or hydrocolloids that is mentioned in lines 397 - 400. A positive control with more sugar (sucrose) would be desirable.

Response: Of course, the hypothesis regarding the presence of pectin in the experimental samples is based on the fact that Cornus mas possesses this hydrocolloid (Pancerzs et al., 2019) and findings from the available literature. Also, it is true that replacement of wheat flour with CCP led to a demonstrable increased content of carbohydrates (Table 3) in the bread loaves as compared not only to the control sample but also among the experimental samples with different CCP additions, except for 5% and 10% (both reaching similar values for this parameter). On the other hand, the volumes of these bread samples were significantly different (Table 5). Therefore, another aspect most probably contributing to the findings has been taken for consideration. Anyway, thank you for the recommendation regarding supplementing the analyses with a positive control with a higher carbohydrate content. We will definitely consider this option in the future.

References:

  1. Pancerz, M., Ptaszek, A., Sofińska, K., Barbasz, J., Szlachcic, P., Kucharek, M., & Łukasiewicz, M. (2019). Colligative and hydrodynamic properties of aqueous solutions of pectin from cornelian cherry and commercial apple pectin. Food Hydrocolloids, 89, 406-415.

Point 3: In lines 409 - 412 an effect on bread viscosity. It was not measured, so it could not be stated.

Response: Revised directly in the manuscript.

Point 4: How the descriptor "overall impression" was defined? This must be specified, it is a descriptor that measures intensity or is evaluating preference. Using the same judges to describe the product and measure preference is not recommended. On the other hand, a lower value could have a higher consumer acceptance, as read in the conclusions (lines 456-457), is it correct?

Response: The descriptor "overall impression" was used to measure intensity and it was defined as: "the balance of the ration of sweet, bitter and sour taste, where min. was minimal balance between the taste parameters , max. was maximal balance between the taste parameters. Lines 456-457 are modified directly in the manuscript.

Point 5: Table 6 shows that the sample with a 10% addition of CCP has the highest pore uniformity. It does not correspond with what is observed in Figure 1.

Response: Edited directly in the manuscript. The Figure was removed based on the recommendations of another reviewer.

Point 6: In Table 1, the comparison of means is unnecessary since they are different products.

Response: Thank you for the opinion. In effect, we included the comparison to provide an idea how these input ingredients differ from each other considering their nutritional values, as CCP replaces wheat flour in the products. Also, this comparison is very helpful for creating a view of hypotheses how the replacement of wheat flour with different concentrations of CCP might influence the measured parameters of the prepared bread loaves.

Point 7: In addition, when the comparison of means is performed, it is usual that the same trend is assigned to the order of the letters. In all tables, there is no order of the letters assigned.

Response: To express a significance in differences of measured parameters between individual samples, we used the standard labelling as it was also used in our previous studies (Valková et al., 2020; Valkova et al., 2021; Valková et al., 2022). Indeed, superscript lowercase letters (e.g. a, b, c) in order from left to right were employed to indicate specific notes, and each table’s first footnote was the superscript a. In this trend, the same letters in each row mean that the values are not significantly different, and different letters in each row indicate the significance. So, the number of superscript lowercase letters depends on the number of various significant differences between the samples.

References:

  1. Valková, V., Ďúranová, H., Štefániková, J., Miškeje, M., Tokár, M., Gabríny, L., ... & Kačániová, M. (2020). Wheat bread with grape seeds micropowder: Impact on dough rheology and bread properties. Applied Rheology, 30(1), 138-150.
  2. Valkova, V., Ďúranová, H., Miškeje, M., Ivanišová, E., Gabriny, L., & Kačániová, M. (2021). Physico-chemical, antioxidant and microbiological characteristics of bread supplemented with 1% grape seed micropowder. Journal of Food & Nutrition Research, 60(1), 9-17.
  3. Valková, V., Ďúranová, H., Havrlentová, M., Ivanišová, E., Mezey, J., Tóthová, Z., ... & Kačániová, M. (2022). Selected Physico-Chemical, Nutritional, Antioxidant and Sensory Properties of Wheat Bread Supplemented with Apple Pomace Powder as a By-Product from Juice Production. Plants, 11(9), 1256.

Reviewer 2 Report

I have now read the article entitled Cornus Mas Powder as a Functional Ingredient: Physical  Properties, Nutritional Value, Bioactive Compounds Composition, and Sensory Attributes of Enhanced Bread Loaves which is interesting by it subject but not from the methodology used. For a such a prestigious journal such as Foods very few analysis and data are presented. In my opinion an article is a complex one in bread making if presents physical-chemical, rheology, dough microstructure and bread quality characteristics. The article presented only data related to the bread quality characteristics and some chemical data related to raw materials used (which are not so fine from my point of view since the wheat flour quality for bread making (if it was of a strong, very good, good quality, etc was not determined). That way this article is not so complex one in my opinion. Only some comments related to the article are mentioned above:

Introduction:

The authors describe different possibilities to use many types of fruits in bakery products and their effect on bread quality. However, they are using in their study cornelian cherries but they do not present any effect of their use (obtained by other authors) on bakery products quality. Please complete the study with more informations on their effect on bakery products according to international literature.

Methods:

Line 129: what warm water means? Also, I do not understand the 2 speeds used: first speed: 131 4 min, 4.0 kW; second speed: 6 min, 8.0 kW. Generally the speeds are reflected in rpm. Also the portion of dough in 250 g is not clear to me – line 134. I know that a bakery products to be named bread according to legislation must have a minimum of 300 g. So from a dough of 250 g it is not possible to obtain bread (maybe a bun). In this case we are not discussing of bread as a final product. The moisture content obtained according to manufacturer's instructions must be described since was not made after a standard procedure (lines 152-154). The same remark for the bread loaves volume analysis (lines 198-201).

Results:

The authors are expressed their sensory results in graph and table. I am not agreeing with both representations due to the fact that are the expression of the same results. Please use only one from the both representations.

Discussions:

Lines 310-318 are not relevant. The data obtained on wheat are common ones. The authors must focus on differences between wheat and cornelian cherries flour. More, the data on wheat flour are irrelevant for this study. Maybe the quality of wheat flour is the most important one. A wet gluten content and it quality (deformation index for example) would be more important for this study.

Author Response

Reviewer #2

I have now read the article entitled Cornus Mas Powder as a Functional Ingredient: Physical Properties, Nutritional Value, Bioactive Compounds Composition, and Sensory Attributes of Enhanced Bread Loaves which is interesting by it subject but not from the methodology used. For a such a prestigious journal such as Foods very few analysis and data are presented. In my opinion an article is a complex one in bread making if presents physical-chemical, rheology, dough microstructure and bread quality characteristics. The article presented only data related to the bread quality characteristics and some chemical data related to raw materials used (which are not so fine from my point of view since the wheat flour quality for bread making (if it was of a strong, very good, good quality, etc was not determined). That way this article is not so complex one in my opinion. Only some comments related to the article are mentioned above:

Response: Thank you for the opinion. We agree that regarding bread production, many parameters including also rheology and microstructure are valuable to determine. However, this study was methodologically designed in line with our previous one (Valková et al., 2022 published in prestigious journal Plants - Basel) dealing with addition of apple pomace powder to bread formulation to create a complexity of our experiments. In such a way, a comprehensive view of impacts of diverse non-traditional fruit additions on bread physico-chemical and nutritional parameters, as well as sensory properties can be drawn.

References:

  1. Valková, V., Ďúranová, H., Havrlentová, M., Ivanišová, E., Mezey, J., Tóthová, Z., ... & Kačániová, M. (2022). Selected Physico-Chemical, Nutritional, Antioxidant and Sensory Properties of Wheat Bread Supplemented with Apple Pomace Powder as a By-Product from Juice Production. Plants, 11(9), 1256.

Introduction:

Point 1: The authors describe different possibilities to use many types of fruits in bakery products and their effect on bread quality. However, they are using in their study cornelian cherries but they do not present any effect of their use (obtained by other authors) on bakery products quality. Please complete the study with more informations on their effect on bakery products according to international literature.

Response: To our best knowledge, the published literature regarding the cornelian cherries related to bakery products are very rare. Now, we have found only one publication (İlyasoğlu et al. 2022; published only recently) dealing with enrichment of cookies with cornelian cherry which was added into “Introduction”. For this reason, our study can be considered as innovative because it provides the first data in this topic.

References:

  1. İlyasoğlu, H., Arslan Burnaz, N., & Arpa Zemzemoğlu, T. E. (2022). Flaxseed and Cornelian cherry: Development of a functional cookie using response surface methodology. Journal of Food Processing and Preservation, 46(11), e16954.  

Methods:

Point 2: Line 129: what warm water means?

Response: The word “warm” was deleted in the manuscript.

Point 3: Also, I do not understand the 2 speeds used: first speed: 131 4 min, 4.0 kW; second speed: 6 min, 8.0 kW. Generally the speeds are reflected in rpm.

Response: Edited directly in the manuscript.

Point 4: Also the portion of dough in 250 g is not clear to me – line 134. I know that a bakery products to be named bread according to legislation must have a minimum of 300 g. So from a dough of 250 g it is not possible to obtain bread (maybe a bun). In this case we are not discussing of bread as a final product.

Response: We used the same bread-making procedure as in the previous study, when we divided the dough into pieces weighing 250 g (Valková et al., 2022). Due to the fact that this is an experimental study, we used lower amounts of dough for the production of bread loaves due to waste elimination. In some studies (Zhou et al., 2019; Xie et al. 2022), lower amounts of dough (80g and 75 g, respectively) were even applied for bread production.

References:

  1. Valková, V., Ďúranová, H., Havrlentová, M., Ivanišová, E., Mezey, J., Tóthová, Z., ... & Kačániová, M. (2022). Selected Physico-Chemical, Nutritional, Antioxidant and Sensory Properties of Wheat Bread Supplemented with Apple Pomace Powder as a By-Product from Juice Production. Plants, 11(9), 1256.
  2. Xie, X., Cai, K., Yuan, Z., Shang, L., & Deng, L. (2022). Effect of Mealworm Powder Substitution on the Properties of High-Gluten Wheat Dough and Bread Based on Different Baking Methods. Foods, 11(24), 4057.
  3. Zhou, J., Yang, H., Qin, X., Hu, X., Liu, G., & Wang, X. (2019). Effect of β-cyclodextrin on the quality of wheat flour dough and prebaked bread. Food Biophysics, 14(2), 173-181.

Point 5: The moisture content obtained according to manufacturer's instructions must be described since was not made after a standard procedure (lines 152-154). The same remark for the bread loaves volume analysis (lines 198-201).

Response: Edited directly in the manuscript.

Results:

Point 6: The authors are expressed their sensory results in graph and table. I am not agreeing with both representations due to the fact that are the expression of the same results. Please use only one from the both representations.

Response: Edited directly in the manuscript.

Discussions:

Point 7: Lines 310-318 are not relevant. The data obtained on wheat are common ones. The authors must focus on differences between wheat and cornelian cherries flour. More, the data on wheat flour are irrelevant for this study. Maybe the quality of wheat flour is the most important one. A wet gluten content and its quality (deformation index for example) would be more important for this study.

Response: Of course, thank you for the analysis proposal of wet gluten content. We will definitely consider it in the future. Here, we provide the comparison of differences between the two input raw materials which was very helpful for creating a view of hypotheses how the replacement of wheat flour with different concentrations of CCP might influence the measured parameters of the prepared bread loaves. This view was consequently useful to draw important conclusions from our results, as well as to prepare “Discussion” in the present paper. Moreover, our previous study (Valková et al., 2022) was also prepared in this way. Hence, a comprehensive picture of impacts of diverse non-traditional fruit additions on bread physico-chemical and nutritional parameters, as well as sensory properties can be provided.   

Round 2

Reviewer 1 Report

Line 413: change tendency by trend

Author Response

Reviewer #1

Point 1: Line 413: change tendency by trend

Response: Edited directly in the manuscript.